# Using health information for community activism: A case study of the movement for change and social justice in South Africa

**Myrna van Pinxteren** [1] *, **Christopher J. Colvin** [2,3], **Sara Cooper** [4]

**1** Department of Medicine, Chronic Disease Initiative for Africa, University of Cape Town, Cape Town, South Africa, **2** Department of Public Health Sciences, University of Virginia, Charlottesville, Virginia, United States of America, **3** Division of Social and Behavioural Sciences, School of Public Health and Family Medicine, University of Cape Town, Cape Town, South Africa, **4** Cochrane South Africa, South African Medical Research Council, Cape Town, South Africa

* myrna.vanpinxteren@uct.ac.za

**Data Availability Statement:** After consultation with A/Prof Christopher J. Colvin, co-author of this paper and Co-PI of the iALARM project, it was confirmed we are not able to share any raw data.

## Abstract

The access to, use, and exchange of health information is crucial when strengthening public health services and improving access to care. However, many health system stakeholders, including community groups are perpetually excluded from accessing and using health information. This is problematic as community groups, themselves end-users of care, are well-positioned to keep the health system accountable, provide feedback on the quality of services, and identify emerging health concerns. Using qualitative, ethnographic methods, this paper investigates different strategies used by the Movement for Change and Social Justice (MCSJ)–a local health activism group–to collect, use and distribute health information to improve health care in Gugulethu, a low-income neighbourhood in Cape Town, South Africa. Through participant observation, shadowing, informal conversations and semi-structured interviews that were analysed using iterative thematic analysis, findings revealed that MCSJ effectively collected, used and exchanged health information to develop short-term health campaigns. To get access to the needed health information, they used innovative strategies, including cultivating allies in the health system, finding safe spaces, and using community brokers to effectively mobilise community members to keep the health system accountable. MCSJ's strategies highlight that stakeholders' engagement with health information is not only a technical exercise, but a complex social process that requires constant negotiation and relationship building. Therefore, to make meaningful improvements to health services and create adaptive and responsive health systems, we need to include community groups as active stakeholders in the health system, provide relevant, up-to-date and locally relevant health information, and facilitate opportunities to socially engage with health information and those who produce it.

## Background

### Health information use and exchange in Sub Saharan Africa

The access to, use, and exchange of health information is crucial to strengthening public health services and improving access to care. While the role of health information is recognized as a

The original iALARM ethics agreement with UCT's HREC, which was obtained in 2014, indicates that we would keep all qualitative research data confidential and would publish interpreted findings only. Queries on data access can be sent to Marc Blockman, head of the UCT Health Sciences HREC. Marc.blockman@uct.ac.za.

**Funding:** This work was supported by a grant from the National Institute of Mental Health and the South African Medical Research Council (R01 MH106600) (MvP, CC) and was part of the South African Social Sciences and HIV programme, supported by a grant from the National Institute of Child Health and Human Development (R24 HD077976) (MvP, CC). The funders had no role in study design, data collection and analysis, decision to publish, or preparation of the manuscript.

**Competing interests:** The authors have declared that no competing interests exist.

**Abbreviations:** ARV, Antiretrovirals; ART, Antiretroviral therapy; CBHIS, Community-based health information system; CBO, Community-based organisation; CHMIS, Community health management information system; CRC, Citizen report card; CSC, Community score cards; CTO, Community treatment observatory; HCP, Healthcare provider; HIS, Health information system; HIV, Human immunodeficiency virus; iALARM, Information to Align Services and Link and Retain Men in the HIV Cascade; iALARM TT, iALARM Task Team; MCSJ, Movement for Change and Social Justice; MSF, Medicines sans frontieres; NGO, Nongovernmental organisation; RHIS, Routine health information system; RMR, Routine management report; SGB, School governing body; TAC, Treatment Action Campaign; TB, Tuberculosis; UCT, University of Cape Town; UNAIDS, United Nations Programme on HIV/AIDS; WCDoH, Western Cape Department of Health; WHO, World Health Organisation.

vital component in the health system and one of the WHO health system building blocks [1,2], not all health system actors have equal access to it, in particular community actors. This is problematic for many reasons, as community actors might desire to use health information to develop evidence-based campaigns, keep the health system accountable, and mobilise and inform the public [1,3,4]. This paper investigates how activism groups and organisations in disadvantaged communities in South Africa can engage more productively with health information to improve health care delivery.

In the health system, the collection, use, and exchange of health information is used by health workers, managers, and policy makers for daily decision-making and to strengthen both the access to and quality of health services [5–7]. This information can be gathered routinely, through patient records, routine management reports (RMR), or demographic data and is ideally stored in Routine Health Information Systems (RHIS) that produce accurate and reliable health information [8]. In reality, however, health information is often over-produced by health actors, who routinely store patient data in (RMR) reports- and management reports. However, this information is under-used, due to fragmentation in the health system, shortages of human and financial resources and ill-defined information needs [9,10].

The available RHISs are also considered technocratic, whereby most health information is only stored in reports and not adequately used in programme development or policy and health systems strengthening efforts [11]. This fragmentation hinders the effective flow of health information within the health system and results in unequal access to health information for health systems actors [12–16]. Additionally, South African data integrity policies are outdated, creating, a continuous challenge to protect private patient information and in turn hinders the effective flow of health information across different levels of the health system [17,18].

To substitute this lack of, and poor access to data, health actors in South Africa frequently use non-routine health information within their daily work, including conversations with patients and colleagues, experiences and observations and various other types of formal and informal tacit knowledge [13,19]. This formal and informal tacit knowledge is obtained through training, skill-building and work experience and considered to be the unwritten memory of an organization, which is hard to measure or quantify [20]. Pamphlets, health education materials and patient experiences are other forms of non-routine health information [21].

Over the past 25 years, South Africa has made tremendous strides in improving access to, and use of information for decision-making in the health system by developing district-based integrated health information systems. However, these systems are mostly inaccessible to non-health actors, such as community activists, NGO's or researchers.

## Information and community health activism

Recently, health information researchers and development agencies based in low-middle income countries attempted to improve the access to information for non-health actors in Kenya and Zambia, mainly through the development of community health management information systems (CHMISs) or community-based health information systems (CBHIS). These are central dashboards where individual health information, demographic data, and disease prevalence data is collected from the community to support decision-making and inform health programmes [22,23]. Information for CBHISs are mostly collected by community health care workers (CHWs) and used to guide evidence-based dialogues between health workers, clinic committees, and the larger community [23]. Using information stored in CBHISs, clinic committees can help set health priorities, make decisions on health

interventions and raise the alarm about emerging health issues [23]. This is important, as community members, who are themselves end-users of care, are well-positioned to keep the health system accountable, provide feedback on the quality of services and identify emerging health concerns which affect themselves and their communities [4,24,25]. Community engagement is also crucial for programme implementation, as well as in monitoring, evaluating and improving the quality of health services [1,3].

In the absence of CBHISs in South Africa, clinic committees are established to monitor the performance of health facilities through ongoing engagement with health workers and community members [26,27]. Clinic committees should have access to health information from facilities, including routine progress reports and routine management reports (RMRs) [28]. There is little literature examining the extent to which clinic committees actually engage with health information. An exception is a case study by Colvin et al., (2011) which found that one clinic committee in a township in Cape Town regularly interacted with health information to raise ongoing health issues in the community [29]. Other strategies used to improve community engagement with health information systems are community monitoring, citizen report cards (CRC), community score cards (CSC) and social audits, which all track the performance of health services [24]. Through these strategies, communities are able to collect their own information to improve services, create access to health services and change the behaviour of health care providers (HCPs) [24,26,30–33].

HIV/AIDS activism campaigns in South Africa also show communities have been collecting, using and distributing health information to improve health and healthcare. Here, HIV activism groups, such as the Treatment Action Campaign (TAC), successfully developed evidence-based campaigns to raise awareness for HIV, destigmatize the disease and advocate for free ARV treatment [34]. In the height of the HIV epidemic in the 1990s and early 2000s, as millions of patients died of AIDS in Africa [35], TAC solicited and leveraged academic information produced in the global North about HIV and available treatments, and used this information to force the government to roll-out ARV treatment in South Africa, improve health services and influence health policies [34,36]. By raising public awareness about issues surrounding the availability, affordability and use of HIV treatments, TAC successfully campaigned for access to ARV treatment [37]. TAC also trained 'expert-patients' to deliver health talks in clinics and to support patients to effectively link and adhere to HIV care. However, aside from TAC's successful treatment literacy programme and ongoing HIV campaigns, there are few other examples showcasing how community activist groups in South Africa have effectively used health information to strengthen health services in low-income settings.

Building on the available literature about community engagement in South Africa and HISs, this paper explores how a grassroots activism group—the Movement for Change and Social Justice (MCSJ)—embraced their role as *active* producers, users and distributers of health information. Specifically, this paper lays out the different strategies used by MCSJ to obtain seemingly unavailable health information, including collecting their own qualitative and quantitative information, to shape health campaigns and effectively inform and mobilise the larger community. Through several examples outlined in the paper [Boxes 1–4], we demonstrated how MCSJ was able to keep the health system accountable to providing equitable, affordable, and accessible health services in Gugulethu, a low-income neighbourhood in Cape Town, South Africa.

MCSJ is a community activism group established in 2016 in Gugulethu, MCSJ develops short- and long-term health campaigns to improve the quality of and access to health services in Gugulethu and neighbouring areas [Boxes 1–4]. MCSJ was founded by Mandla Majola, an experienced HIV activist from Gugulethu. Through his activism background and role as community engagement manager in the iALARM study–a longitudinal study aiming to improve

Box 1. MCSJ's dentistry campaign

### 1: Dentistry campaign

One of MCSJ's early campaigns was to reduce the waiting times at a local dentistry. Originally, there was only one dentist at the clinic, who would operate services with a few students. The clinic was also open only on a Wednesday and could only help 30 people per day. This led to queuing from as early as 4am and patients paying a ticket for ZAR30 (USD 1,5) to secure a place to be helped. When MCSJ members became aware of this problem, they monitored the activities at the clinic and collected 74 testimonies from patients waiting in line for the dentistry. After collecting this data, they arranged a meeting with the clinic manager, primary care manager, and dentist, where they presented collected information and compared them with the clinic statistics. Even though the provided statistics showed that the clinic was assisting more patients than other dentistry's in the area, MCSJ argued to the clinic team that public health is a human right and in South Africa should be provided free of charge. After consideration, a locum dentist was appointed and together with the dentistry team, was able to resolve the queues at the clinic.

the testing and linkage of men to HIV services Mandla recognised the longstanding health issues in the community, including the high prevalence of HIV and non-communicable diseases, and lack of activist efforts [38,39]. Worried about the situation, he shared his concerns with other activists in Gugulethu, and collectively, they set-up the Movement for Change and Social Justice (MCSJ). MCSJ initially started with 15–20 committed activists, but has grown significantly over the past five years, establishing branches (offices) in other low-income neighbourhoods in Cape Town, including Nyanga, KTC, Kanana and Phillipi.

Box 2. MCSJ's Condoms in schools campaign

### 2: Condom in schools campaign

HIV-infection among adolescents and teenage pregnancy is a longstanding problem in South Africa. This leads to school drop-out (especially for girls) and increased health risks. To tackle this problem, MCSJ developed a campaign to supply free condoms (male and female) in high schools in Gugulethu. They worked together with schoolteachers to convince the School Governing Body (SGB)–the board responsible for policies in schools–to allow students access to condoms on the school premises, and developed pamphlets in English and isiXhosa with recent statistics on the implications of teenage pregnancy which were distributed in the community. MCSJ also worked with the Treatment Action Campaign to produce a documentary stressing the importance of providing condoms in schools [https://www.youtube.com/watch?v=Kz1OUDMM56M]. This documentary was shown in high schools in the community, accompanied by a lesson in sexual and reproductive health (SRH) delivered by Rachael Gibson, a SHR specialist from the US linked to the iALARM project.

## Box 3. iALARM Taks Team and Research Indabas

### 3: iALARM Task Team & Research Indabas

Since early 2017, MCSJ members have regularly attended iALARM Task Team (TT) meetings, whereby health workers, researchers and NGO's come together to speak about men and HIV. At these meetings, iALARM researchers would present HIV reports with locally relevant information on men's trajectory in the HIV cascade in Gugulethu. Over time, the TT provided opportunities for participants to request information and to share their own ideas on how to improve men's HIV testing and linkage to care. MCSJ's role in the TT led to several follow-up projects, including a clinic poster to make health facilities more male-friendly, a dedicated Men's Forum where men from Gugulethu could come together to speak about HIV, other health conditions, and changing pervasive gender norms. Together with the iALARM Team, MCSJ also organised two successful Research Indaba's (meetings) in 2018 and 2019, where researchers presented findings from the iALARM and other HIV research projects to between 500 and 1000 community members and engaged with them in plenary and break-out sessions (https://ialarm.org.za/events/).

## Box 4. Triage in a local clinic

### 4: Triage in a local clinic

Since their commencement in 2016, MCSJ became well-known in Gugulethu, due to their ongoing campaigns in health facilities and engagement with the iALARM Team. Their wide networks in the community and knowledge of the health system inspired a family physician at Gugulethu clinic, Dr. Morgan, to approach MCSJ to ask for assistance with regulating overcrowding of the ER in the weekend. In a first meeting, Dr Morgan showed the statistics from the facility, depicting a large number of patients flocking to the ER during evenings and weekends. This led to long waiting times for patients that needed to be treated for minor illness, as life-threatening injuries would be prioritised. The frustration among patients was caused, at least in part, by the lack of knowledge about the hospitals triage system. To educate patients, clinic staff developed pamphlets in isiXhosa to explain the triage system according to colours (red, orange and yellow in order of priority) and urged people to pick-up their medication on time, during weekdays. The clinic also provided materials to promote child health services and the extended opening times, which were distributed by MCSJ in the community. Dr Morgan also gave a well-received presentation about the clinic during the first Research Indaba in 2018 (see Box [3]).

## Methodology

### Ethics statement

This research involved human participants. Ethical approval has been granted by the Human Research Ethics Committee (HREC) at the University of Cape Town (318/2017). The Western Cape Department of Health (WCDoH) and the City of Cape Town (802/2014) gave access to conduct research in facilities within the Klipfontein Sub-district.

### Study design

This study is part of a larger ethnographic research conducted between March 2016 and September 2018 that explored the role of health information in the low-income community of Gugulethu, Cape Town, South Africa. This study design was fitting, as it allowed the researcher to participate in local realities of participants and understand the social and historical contexts in which health information is accessed, used and exchanged [40–42].

Gugulethu is a low-income peri-urban community located 15kms from Cape Town. The township is inhabited by approximately 1000000 people, mostly black, isiXhosa speaking South Africans. As with many other townships in South Africa, Gugulethu was established after the implementation of the Native Urban Areas Act in 1950. Similar to elsewhere in the country, Gugulethu experiences socio-economic challenges and has a quadruple burden of disease, including high rates of HIV/AIDS, tuberculosis (TB), high maternal and child mortality as well as non-communicable diseases; violence and injuries. There are three primary care facilities in the neighbourhood, which offer care for communicable and non-communicable diseases. Additionally several NGOs offer health education training and social support programmes. Gugulethu and the larger Klipfontein Sub-district were chosen as research sites for the iALARM study. iALARM's primary focus was improving the communication and coordination of services between the NY3 community health clinic in Gugulethu and the Men's Wellness Centre, an initiative from Sonke Gender Justice (SGJ), located on the grounds of the clinic [12,38,39].

For this paper, the first author, Myrna van Pinxteren (PhD) (MvP), worked closely with 10 key-informants, who were introduced to her through Mandla Majola, one of the founders of MCSJ. All key-informants, 6 men and 4 women, worked as HIV activists or at community NGO's (including Sonke Gender Justice, Grassroots Soccer and The Parent Centre) and were directly linked to the social movement. This paper is one of three linked case studies aiming to understand how research, community activists and health workers accessed and used health information to improve health services in Gugulethu.

### Data collection

As a primary researcher, MvP collected data for this paper using a variety of ethnographic methods, which include participant observation, shadowing, informal conversations and semi-structured interviews [42,43]. She also observed MCSJ team meetings and engaged in their ongoing activities and health campaigns [Boxes 1–4]. All data was collected face-to-face. MvP established a close relationship with MCSJ during her 18-month period of intensive fieldwork provided an opportunity for frequent conversations, reflections and discussions which were iterative in nature [44–46]. MvP travelled to Gugulethu 2 or 3 times a week, depending on ongoing MCSJ activities. Findings from fieldnotes were discussed with key-informants to ensure data validity. Data was also collected from informal conversations with key-informants, informed by queries and questions deriving from fieldnotes [45]. Key areas of interests were MCSJs relationship with health providers and researchers, strategies used to access health

information and role in the larger iALARM project. All interactions between the researcher and key-informants were in English and conversations were audio-recorded and transcribed by MvP.

As this is an ethnographic study, issues around reflexivity and subjectivity were considered [47–49]. As a non-South African working in low-income contexts such as Gugulethu, MvP's observations may have influenced her views on MCSJs activities, as she had close relationships with her key-informants. To navigate this space as an 'outsider' emerged in researcher the social movement, MvP had continuous conversations with SC and CC about her positionality and regularly shared her observations and concerns. MvP also travelled to Gugulethu 2 or 3 times a week for 18 months, allowing her enough time to reflect on her findings [46].

## Data analysis

All fieldnotes, observations, meeting minutes and conversations were written in English and stored on a password protected computer. This data was analysed in Nvivo, using an iterative thematic analysis approach used to 'identify, analyse and report on patterns (themes) within the data', whilst keeping the context in which the data was collected [50]. This was a systematic and inductive process which was largely conducted during data collection, using grounded theory [51]. Firstly, fieldnotes and interview transcripts were coded by MvP who identified themes. These themes were verified across different fieldnotes and linked to the research questions. Later, categories and sub-themes were added. Then, MvP consulted with SC and CC for further refinement of generated themes. These included MCSJ's capacity to create allies, to gain access to health information and the role of 'safe spaces' for effective health information sharing. Generated themes were further crystallized after repeated discussion between the authors and feedback from key-informants in the project [52].

## Ethical considerations

Ethical approval for the research was obtained from the Human Research Ethics Committee at the University of Cape Town (380/2017). The Western Cape Department of health (WDoH) and the City of Cape Town (802/2014) gave access to conduct research in health facilities within the Klipfontein sub-district. As this research project was nested in the larger iALARM study and developed as an ongoing evaluation of the project, consent forms from key-informants were obtained by iALARM researchers. All participants were informed about the objectives of the conducted research and linked iALARM study and at every point of contact, MvP explained her role as a researcher in the group. This research project was designed using an longitudinal ethnographic study design and aligned with disciplinary convention, MvP conducted research in larger meetings or public settings. Therefore, written consent could not be always be obtained [43,45,46,53]. During her time in the field, MvP created strong relationships with key-informants in communities and colleagues within the project.

As these key-informants made important contributions to this paper, it is appropriate and respectful to share their names in this and other outputs of her PhD study [43,53,54]. MvP received written consent from all named key-informants in this paper.

## Findings

Since its inception, the Movement for Change and Social Justice (MCSJ) has orchestrated various short-term health campaigns to mobilize communities and improve health services. Information played a central role in these campaigns, and MCSJ employed creative various strategies to access, collect, use and/or distribute health information. The 4 textboxes [Boxes 1–4] outlines the various MCSJ campaigns and type of data used. Using these examples, we

unpack the innovative strategies used by MCSJ, including their ability to cultivate allies in the health system, finding safe spaces, and using community brokers to effectively mobilise community members to keep the health system accountable.

## Cultivate your allies

MCSJ used various sources of health information for their campaigns, including self-collected testimonials and academic information. Most information was not readily accessible to the activists, urging them cultivate allies by working closely with academic researchers and health system stakeholders in order to obtain the required information. For the 'condom in schools' campaign, MCSJ aimed to design an evidence-based pamphlet containing statistics on teenage pregnancy and HIV infection rates, designed using a academic framework. According to the activists, this academic data would convince the school governing bodies (SGBs)–the boards that decide on school policies in South Africa—on the need to distribute condoms in school.

However, as MCSJ did not have this information, they approached iALARM researchers with whom they co-designed a 1-page pamphlet in English, which outlined the issues at hand and included some colourful and easy to read graphics [S1 Text]. Once finalised, MCSJ connected with local church leaders who had contact with SGB members to deliver the pamphlets and initiated a dialogue on the necessity of providing condoms in schools. When this proved ineffective, MCSJ members teamed up with allies from TAC in neighbouring Khayelitsha. Together, they developed a 15-minute documentary to create awareness about pregnancy in schools and advocate for the need to distribute condoms to reduce school dropouts in the province.

For the dentist campaign [Box 1], MCSJ activists collected testimonials from patients waiting in the clinic queue to convince health managers in Gugulethu to hire an extra dentist to reduce the waiting times at the dental clinic. Although MCSJ assembled the information themselves through the self-census method, they heavily relied on the relationship between Mandla and the clinic manager to present these testimonials to the dentist team and primary care manager. Mandla and the clinic manager grew up together and met regularly to discuss ongoing research activities in Gugulethu. When Mandla approached him for a meeting, he was happy to facilitate it, as long as MCSJ did not come and *'toi toi (protest) in front of his clinic'*. Within a week, the facility manager arranged a meeting between MCSJ, the dentist team and the primary care manager. At this meeting, activists were given an opportunity to present their self-collected health information to the group, compare these with the statistics provided by the clinic, and brainstorm about possible solutions to improve the services at the dentist.

This alliance between MCSJ and health system actors also proved reciprocal, as over time, MCSJ also received routine clinic data (RMR) from the family physician in a Gugulethu clinic [Box 4]. RMR data is normally unavailable to the public, but was shared during a meeting with MCSJ called by the family physician who needed assistance communicating triage regulations to community members. *"Patients might have a cough, backpain or flu, but still need to wait for more than six or seven hours to get it solved, or sometimes they are sent away completely. This increases the tension in the clinic and if there is annoyance among those patients, they might not come back at all, or resort in violence, which is really dangerous for me and my staff" (Family physician, Community Health Clinic).*

Receiving this data and being confronted with the urgency of the situation, MCSJ agreed to educate community members about the triage system, through distributing pamphlets and inviting the physician to share her concerns large community meeting in Gugulethu (Research Indaba) [Box 3]. Speaking at this meeting, she was able to reach out to more than 500 people simultaneously.

Effectively using personal relationships in the community, health system and academic networks to get access to health information, MCSJ both presented their self-collected information and distribute health information. The also mobilised communities and raised awareness on emerging health issues. These relationships with health system stakeholders proved to be reciprocal, as MCSJ assisted with finding opportunities to share messages from health actors to the larger community of Gugulethu.

## Find safe spaces

Aside from cultivating allies to access and distribute health information, MCSJ participated in 'safe spaces' with health actors and academics, creating an opportunity to actively engage with health information and discuss its possibilities and limitations. This is unique, as health information, and routine clinic data specifically, often sits in reports, but not openly shared among health and non-health actors. These 'safe spaces' are not provided in South African health settings, or do not are considered hostile and contentious for all parties involved Health workers might feel blamed for not providing timely services for patients, having hostile attitudes or the stock-out of medication, even though this is largely out of their control [55,56]. Patients or community representatives might be accused of irresponsible health decisions or non-adherence, without getting the opportunity to unpack this behaviour [55–57]. Presenting health information during these already uncomfortable conversations between community and health systems stakeholders, can potentially increase hostility instead of starting constructive dialogues on how to collectively improve health services. Therefore, building trust between participants and compassionate discussion leaders are key to facilitate discussion, rather than using the information to pinpoint persistent problems in the health system or problematic patient behaviour.

To improve linkage of men to HIV care and promote the communication and information sharing between health actors and the larger community, the iALARM Task Team (TT) invited MCSJ members and HCPs to discuss HIV reports and other information in a bimonthly meeting hosted by researchers from the University of Cape Town (UCT). During these meetings, MCSJ received up-to-date HIV information in which was used in their campaigns. They also requested information from TT members about health issues affecting the community, including how to manage non-communicable diseases (NCDs) or how to sensitize community members about gender based violence (GBV). [Box 3] [12]. Additionally, MCSJ received information about the impact of HIV in their community and participated in group discussions with different health stakeholders.

In this shared space, MCSJ members and health actors took the opportunity to question each other, the provided information or request clarifications. After reading the first HIV report shared by the iALARM team, one MCSJ member commented: *"Some men think that the clinical environment is not conducive for men. If you go to the clinic, all the charts represent women. The clinic is just not well catered for men" (MCSJ member, Gugulethu)*. This comment sparked a lively discussion in the iALARM TT and marked the start of the development of a male poster aiming to make clinics and community spaces more male-friendly. Organising male dialogues was another initiative that derived from conversations in the iALARM TT. *"We need to have one-on-one conversations with men. This report can help us with that. Because this [data] is what is been collected in our community. So we can share this this information with others" (NGO representative, Gugulethu)*.

Through this engagement, MCSJ members learnt about the functioning of the health system, barriers to HIV testing, the impact of HIV on co-morbidities and other chronic diseases and were updated about new interventions and research in the community. Keen to share

their knowledge with others, MCSJ requested to distribute some of health information to a wider audience during the annual Research Indabas organised by MCSJ and the iALARM TT in 2018 and 2019. During these community meetings, Gugulethu residents directly engaged with researchers and students about ongoing health issues and collectively brainstormed about new avenues for research.

Aside from participating in 'safe spaces' organised by the iALARM team, MCSJ activists also created opportunities for 'safer' engagement with health actors by organising a meeting between the dentist team, the primary care manager, clinic manager and health activists. Rather than mobilizing large community crowds and giving demands, MCSJ opted for a small meeting with the clinic staff to showcase the testimonials from patients and initiate a conversation about improving the dentist services [Box 1]. This engagement initiated brainstorming about solutions to reduce the waiting times outside the clinic and provided an opportunity for the dentistry team to present their own evidence, showing that the dentist operated on full capacity and assisted more patients than surrounding clinics. *"If you look here"*, the facility manager said, pointing at the target sheet, *"you can see that that this dental clinic has treated 12,000 patients in the past 18 months, compared to second dentistry on the list, that only treated 7,700 people in the same time period. The other clinics perform worse" (Facility manager, Gugulethu Community Health Clinic).*

This meeting could have led to contention between activists and the dentist team, but the intimate way of engaging with health actors provided an amicable environment where both parties presented their information and discussed the most effective ways forward. This safer engagement presented MCSJ with an opportunity to push for action and the dentist team to present alternative perspectives. In turn, this resulted in a unique encounter that would not be realizable when protesting. Data presented by the dentist furthermore highlighted that the clinic assisted patients from other sub-districts, increasing the workload in the clinic, something unbeknown to MCSJ members. Reflecting on both MCSJ's testimonials and the clinic data, the primary manager appointed an extra, locum dentist to relieve the work pressure for the existing dentist team and assist more patients.

## Produce your own health information

MCSJ successfully used different strategies to gain access to seemingly unavailable health information by cultivating allies or participating in safe spaces, but in some cases, MCSJ produced their own evidence, employing similar techniques to those used by HIV activists, including creating testimonials to get buy-in for campaigns. In the early 2000s, TAC used testimonials to raise public awareness to improve the availability and affordability of ARV treatments [37]. These testimonials are personal stories of people living with HIV (PLHIV) who share their experiences of their illness and the positive impact of received ART on their personal health wellbeing of their loved ones. MCSJ used similar strategies, sharing stories from PLWH in Gugulethu to convince community members to link to care or trying to inform health workers how HIV services could be improved. Phumzile, one of the MCSJ activists, shared his testimony in one of the iALARM TT meetings.

> *"I am here as a man, a Gugulethu man. And I am shocked about the stories and the numbers that I hear in these iALARM meetings. We need to put in more effort to get men tested. There is a lot of activism is this community, we just do not see it. . .. We are still struggling here in Gugulethu. We really need to start with our facilities and speak to the staff. That they need to put men on the agenda. Speak to men and about men. And we need to start organising men's dialogues and initiate male adherence clubs that are led by men like me, HIV+ men. . . We, as*

*men, should also be present in the waiting rooms of the clinics and share our stories. Show that we are not afraid to disclose, that we live positively. . .. We should speak to men and tell them that we support them. That we have the information and are happy to share. . . We need to reach out to men with living testimonies, just like me" (iALARM field-coordinator, Gugulethu).*

Here, Phumzile emphasized the ongoing stigma towards men in health facilities and made a plea for the improvement of support structures for HIV+ men, a largely neglected group in the HIV response [58,59]. Phumzile's account initiated follow-up projects taken on by the iALARM TT, including the abovementioned poster campaign and a series of discussions on men's health in local clinics. During a recent COVID-19 awareness campaign in Gugulethu, MCSJ also used testimonials, going door-to-door sharing accounts of people who contracted and survived COVID-19. In this way, testimonials were used to educate people about the pandemic, to alleviate fears about testing and self-isolation and to mitigate stigma [60].

MCSJ also collected their own quantitative data from patients queuing the dentist, using self-census information–a method that has been used effectively by the Social Justice Coalition to demand improved social housing, infrastructure, and health services in South African townships [61]. Giving further context, Mandla explained that:

*"You need to get permission of the patient, and if he or she is afraid of victimisation we need to be keep them anonymous. But when they agree to give their ID number, exact names and addresses, it matters. To show, as an activist, that you are not falsifying and coming up with ghost people. If we want to keep the health system accountable, we need to be truthful and honest at all times" (iALARM field coordinator/MCSJ member, Gugulethu).*

Using self-census as a strategy proved effective when the health information was presented to the clinic manager and dentist team. Sharing the 74 accounts from patients, MCSJ showed their engagement with patients and effectively represented the community. This strategy created trust between MCSJ and the larger community, and convinced the dentist team and clinic manager to think 'outside the box' to find a suitable solution.

## Embedding health campaigns in the local context

A final strategy MCSJ developed to improve information usage and sharing information was to embed their activism efforts in the local context of Gugulethu. MCSJ worked closely with community 'brokers', including religious groups, community leaders and local doctors to collect and distribute relevant health information and mobilize community groups to support ongoing campaigns. Mobilizing local teachers proved beneficial during MCSJ's condom campaign [Box 2], as educators would allow MCSJ to educate their learners about sexual and reproductive health, give pamphlets to parents and lobby with SGBs to stress the importance of condom distribution in schools.

To expand the reach of their campaigns, MCSJ also translated technical health information into understandable and locally relevant information. By being involved in the iALARM TT, MCSJ members had access to HIV reports and used this information to develop their own pamphlets or to guide discussions about men's health and access to care. In turn, these activities led to the development of a male friendly poster, designed by local artists, and written in isiXhosa. To better inform the community about men's engagement in HIV care, MCSJ actively shared minutes from iALARM TT meetings to street committees and members of grassroots civic organisations, who in turn became street level supporters for MCSJ campaigns

[62]. In April 2020, MCSJs connection with community leaders and civic organisations were key in the development and implementation of a rapid COVID-19 response where activists—despite strict lockdown regulations–distributed information pamphlets in the township, handed out masks and assisted the WCDoH to track patients in their community [60].

MCSJ's links to academic networks and access to up to date health information also empowered other grassroots organisations to become part of the iALARM TT. By engaging with the iALARM TT, organisations could put in their own requests for information and co-develop follow-up projects. For example, a local soccer coach requested assistance from researchers to facilitate gender- and health training to improve the communication between coaches and young boys about mental and physical health challenges, which was delivered in early 2020. The engagement of community members in the iALARM TT also led to setting up a dedicated Men's Forum, where men from Gugulethu would discuss issues affecting their health, including HIV, chronic diseases and substance abuse [63].

Embedding campaigns into local contexts and working actively with community brokers strengthened MCSJ's presence in the community, successfully linked health system actors to the larger community, and encouraged further engagement and health information sharing in Gugulethu.

## Discussion

Findings in this study demonstrate how a local activism group in Cape Town successfully collected, produced and exchanged health information to inform evidence-based campaigns and strengthen health service delivery. Effectively engaging with health information also improved community engagement, health information sharing, and fostered health system responsiveness [1,4,38,64]. Critically, this paper emphasizes that stakeholders' engagement with health information is not only a technical exercise, but a complex social process that requires constant negotiation and relationship building.

Considering the use and exchange of health information by community groups as a social process offers an alternative to the dominant lens for understanding the effectiveness of health information systems. This lens extends the critique from other scholars that collecting health information is a largely technocratic exercise, which they argue leads to only using routine information for reporting purposes, or storing it away in places inaccessible to most actors in the health system [6,11,23,65,66]. In several LMICs, efforts have been made to include community stakeholders more actively in the production and exchange of health information by developing community-based health information systems (CBHISs) [23,66]. Although these are important health information strengthening projects, such interventions mainly focus on the provision of health information to set health priorities, without necessarily encouraging community groups to engage with information to develop their own campaigns. Furthermore, CBHISs are largely designed to give community members access to information produced by the health system. As such, community groups are still largely considered to be end-users of care, not active stakeholders who have the capacity to collect, produce and distribute health information themselves, as shown in the paper. CHBISs also neglect to see health information use and exchange as a social process, whereby stakeholders are constantly negotiating to get access to the required information, are discussing the relevance of this information within safe spaces and if needed, collecting information themselves [23]. Therefore, just designing a system that stores information for community representatives to access, hereby ignoring the social and political processes, will not likely improve health information use and exchange across different levels of the health system.

MCSJ used various strategies to keep health system actors accountable to providing a high standard of care, such as collecting testimonials through self-census. These strategies have

been similarly described in the literature as Community Treatment Observatories (CTO). In short, CTO is a community-monitoring mechanism that collects information on the stock-out of ARVs and monitors trends on the cascade for care [67]. When used well, CTOs can bring health partners and civil society together to discuss gaps in service provision, find hard-to-reach patient groups and streamline health services [67]. In Sierra Leone, a CTO was set up by UNAIDS and local partners to generate information to support and monitor HIV programmes and promote patient-led HIV-advocacy in the country. Similarly, in Uganda, developing a CTO proved to be an effective tool to upgrade the services at a community health unit as it encouraged patients to develop their own plans to improve services and keep health staff accountable for providing high quality care, which in turn improved the utilization of services and health outcomes [68]. While both the CTO programmes in Sierra Leone and Uganda were praised for their simplicity, involvement of community-members in data-sharing and positive improvement to delivered health services, they were led by international health organisations and researchers who set priorities and targets. Even though Bjorkman & Svensson propose the Ugandan CTO to be ideally community-led and sustained, they only describe the role of CBO's as facilitators of meetings, but do not lay-out a long-term vision of CBO involvement in health system strengthening in Uganda [68].

In contrast, MCSJ's efforts to do community monitoring were driven by local activists, setting their own agenda and priorities to improve the health services, rather than tackling issues proposed by interventionists. By building relationships with both HCPs and researchers and activating local health information, MCSJ was able to put their priorities firmly on research agenda and effectively monitor the health services provided in Gugulethu. Therefore, in contrast to the literature exploring the use of CBHISs and CTO's to improve engagement and promote the use of health information among community groups, MCSJ advocacy efforts were driven from the ground up, formalised by experienced activists who knew the longstanding health issues in the community and had the right network to collect, use and exchange locally relevant health information for their own activism efforts. Additionally, by working closely with researchers, MCSJ members got the opportunity to critically engage with health information and could share their knowledge and experiences with others in the community.

When developing evidence-based health campaigns, one of MCSJ's strengths was their ability to embed both the received and collected information in the local context, which triggered the interest of community members and local opinion-leaders. This interest was caused by a 'hunger for information' and absence of up-to-date health information in Gugulethu [38,60]. Responding to this 'hunger for information', MCSJ created opportunities to bring information to the community and link health actors to community groups, especially during the Research Indaba's organised in 2018 and 2019 [Box 3]. By doing this, MCSJ provided localized health information to Gugulethu, and mobilized the community to effectively ask for more. Again, the process of creating engagement was largely social, using health information as a tool, a common denominator between two groups of people who otherwise would not operate in the same space. By empowering community members to demand health information, MCSJ embraced strategies from the Treatment Action Campaign (TAC) and MSF in the HIV-epidemic, as they facilitated the mediation and communication between health 'experts' and the public [24,34,35,69].

The findings from this paper provide insights into how community health stakeholders might be more effectively included in the larger health system. When community groups have more access to health information and can use this information meaningfully, this can lead to a more engaged community, more responsive health system and ultimately, better health outcomes for patients. Therefore, perpetually excluding community partners from engaging with health information is a missed opportunity, as community groups are often well-informed,

liaise directly with local residents and understand health programmes. This makes them excellent allies withing the health system, whilst simultaneously keeping the health system accountable [1,4,38].

Knowing this, there are three cross cutting lessons we can draw from MCSJ's experiences. Firstly, the paper shows that the engagement with health information by community groups is a social process, not a technical exercise, wherein stakeholders and their relationships play a crucial role in accessing, providing and distributing health information. Secondly, community activist groups are well-positioned to reach vulnerable populations, anticipate forthcoming health challenges and motivate residents to participate in health activities [3,70]. Community groups such as MCSJ are also capable of facilitating dialogues across different levels of the health system [3]. Third, within the overstretched South African health system, primary care services shift from the clinic to the community, relying heavily on the support and expertise of community-based health workers. With this shift, community stakeholders are becoming more central to care, increasing the need for them to have access to and engage with health information. This paper provides lessons on how the provision of and access to health information for community stakeholders could be facilitated. When community stakeholders have access to up-to-date health information plus the opportunity to socially engage with both the health information and those who produce it, they can make decisions about care, inform fellow residents about persistent health issues and update community members about programmes which are continuously restructured. In line with Guenther et al. (2014), we therefore argue that there is a need to design new health information systems whereby high quality, locally relevant health information is provided for and produced by community actors [71]. When designing these systems, the process of using and exchanging should be continuously considered. Only when we actively include community groups and consider health information use and exchange as a social process rather than a technical exercise, can we make meaningful improvements to health services, increase the engagement of community members and create adaptive and responsive health systems.

## Conclusion

This study explored the collection, use and exchange of health information among community activists to strengthen health service delivery in low-income settings in South Africa. By using unique strategies, MCSJ's experience highlights that engagement with health information is a complex social process, and community stakeholders are valuable players within this process, as they are capable of linking health systems actors to patients. Therefore, when redesigning HISs to strengthen community based and patient-centred care, community groups such as MCSJ should be considered as critical partners.

## Supporting information

**S1 Checklist. COREQ_checklist.**
(PDF)

**S1 Text. Appendix A.**
(DOCX)

## Acknowledgments

The authors would like to acknowledge Mr Mandla Majola and all members of the Movement for Change and Social Justice (MCSJ) for participating in this study and allowing the researchers to be part of their ongoing advocacy efforts.

## Author Contributions

**Conceptualization:** Myrna van Pinxteren.

**Data curation:** Myrna van Pinxteren.

**Formal analysis:** Myrna van Pinxteren.

**Funding acquisition:** Christopher J. Colvin.

**Methodology:** Myrna van Pinxteren.

**Project administration:** Myrna van Pinxteren.

**Supervision:** Christopher J. Colvin, Sara Cooper.

**Writing – original draft:** Myrna van Pinxteren.

**Writing – review & editing:** Christopher J. Colvin, Sara Cooper.

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
