## [Decision Letter · Decision Letter 0]

21 Feb 2022

PGPH-D-21-00746

Using health information for community activism: A case study of the Movement for Change and Social Justice in South Africa.

Dear Dr. van Pinxteren,

Thank you for submitting your manuscript to PLOS Global Public Health. After careful consideration, we feel that it has merit but does not fully meet PLOS Global Public Health’s publication criteria as it currently stands. Therefore, we invite you to submit a revised version of the manuscript that addresses the points raised during the review process. 

We have now received feedback from our reviewers (see below) and, as you will note, they are enthusiastic about the originality of the research. However they also offer a number of useful comments/suggestions, particularly around the structuring of the paper, data usage/ data analysis and the reporting of the research. We would be pleased to receive a revised manuscript from you, which addresses all the reviewers' comments. Along with the many helpful suggestions from the reviewers, please consider carefully how you report the Ethics procedures/ ethical issues for this research, the various data sources utilised, and the analysis of the data which was conducted on each.

We look forward to receiving your revised manuscript.

Kind regards,

Isabelle Uny

Academic Editor

Journal Requirements:

1. Please provide additional details regarding participant consent. In the ethics statement in the Methods and online submission information, please ensure that you have specified whether consent was informed.

2. Please provide additional details regarding participant consent. In the ethics statement in the Methods and online submission information, please describe how verbal consent was documented and witnessed when verbal consent was obtained instead of written consent.

3. Your manuscript is missing the following sections: Results. Please ensure these are present, and in the correct order, and that any references to subheadings in your main text are correct. An outline of the required sections can be consulted in our submission guidelines here: 

https://journals.plos.org/globalpublichealth/s/submission-guidelines#loc-parts-of-a-submission

4. Please update the completed 'Competing Interests' statement, including any COIs declared by your co-authors. If you have no competing interests to declare, please state "The authors have declared that no competing interests exist".

5. In the online submission form, you indicated that "Data can be shared by approaching the authors and will only be possible within the boundaries imposed by informed consent and a data-sharing agreement with the University of Cape Town.". All PLOS journals now require all data underlying the findings described in their manuscript to be freely available to other researchers, either 1. In a public repository, 2. Within the manuscript itself, or 3. Uploaded as supplementary information.

6. Please amend your detailed Financial Disclosure statement. This is published with the article, therefore should be completed in full sentences and contain the exact wording you wish to be published.

ii). State the initials, alongside each funding source, of each author to receive each grant.

iii). State what role the funders took in the study. If the funders had no role in your study, please state: “The funders had no role in study design, data collection and analysis, decision to publish, or preparation of the manuscript.”

Additional Editor Comments (if provided):

Thank you for submitting your above referenced manuscript to PLOS Global Public Health.

We have now received feedback from our reviewers (see below) and, as you will note, they are enthusiastic about the originality of the research. However they also offer a number of useful comments, particularly around the structure of the paper, data usage/ data analysis and the reporting of the research.

We would be pleased to receive a revised manuscript from you, which addresses all the reviewers' comments. Along with the many helpful suggestions from the reviewers, please consider carefully how you report the Ethics procedures/ ethical issues for this research, the various data sources utilised, and the analysis of the data which was conducted on each.

Reviewers' comments:

Reviewer's Responses to Questions

**Comments to the Author**

1. Does this manuscript meet PLOS Global Public Health’s publication criteria? Is the manuscript technically sound, and do the data support the conclusions? The manuscript must describe methodologically and ethically rigorous research with conclusions that are appropriately drawn based on the data presented.

Reviewer #1: Partly

Reviewer #2: Partly

2. Has the statistical analysis been performed appropriately and rigorously?

Reviewer #1: I don't know

Reviewer #2: N/A

3. Have the authors made all data underlying the findings in their manuscript fully available (please refer to the Data Availability Statement at the start of the manuscript PDF file)?

Reviewer #1: Yes

Reviewer #2: No

4. Is the manuscript presented in an intelligible fashion and written in standard English?

Reviewer #1: No

Reviewer #2: Yes

5. Review Comments to the Author

Reviewer #1: Comments on content:

Overall comment

This paper presents original research in a well-structured manner that could have significant impact for highlighting the role of community groups in collecting health information and running healthcare campaigns. I would suggest the following changes to be made to fit the PLOS guidelines and become an excellent paper. My most important comment is regarding the lack of information on coding and transcription processes in the methodology.

Background

Comment 1

I would recommend a paragraph or so discussing the ethics of recording and sharing health data.

Comment 2

I would recommend an initial breakdown of the different types of health data that you mention in your findings and a brief discussion on pros/cons of each type. You talk a lot about alternative types of health data in your findings/discussion but only briefly introduce informal, non-routine health data in the paragraph starting on line 74.

Comment 3

Could you elaborate on the point of “over-produced and under-used” made in line 68?

Methodology

Comment 4

I would personally suggest including the first section on MCSJ and Gugulethu in the background section. Set the scene earlier on.

Comment 5

In the paragraph starting on line 157 you detail information about Gugulethu. Could you elaborate on why the information you mentioned is relevant? What does this mean for your findings? Use this space to justify why you researched this initiative and why this area. What benefit will it bring to conduct research here?

Comment 6

The methodology is missing some very significant information. What type of coding approach was used? How were the interviews recorded? Were they transcribed? You mention transcripts briefly in the ethical considerations, but don’t describe this earlier on. Make it precisely clear how you collected and analysed data.

Findings

Comment 7

This section could be strengthened with more use of (anonymised) quotes. Make use of the rich qualitative data available to you here!

Comment 8

You introduce a lot of new concepts and make statements like “RMR is normally unavailable to the public” (line 245) that are not backed by references. These should be introduced and discussed in the introduction. How are these data collected, are they accessible, how could they help community groups, etc.

Discussion

Comment 9

The discussion is very good. Your idea of community groups engaging in a complex social process and portray them as actors engaging in multiple elements of health information sharing is very interesting. I feel you could highlight this point a little more strongly.

Comment 10

I would recommend a separate conclusion section to make your main arguments and recommendations clear to the users.

Grammatical and structural errors:

As per PLOS Guidelines, I have noted the following grammatical errors and typos.

Comment 11

As the rest of the article is written in British English, I have assumed to note the lack of “Oxford commas”.

Oxford comma missing on lines 30, 34, 48, 54, 59, 61, 63, 64, 66, 76, 85, 88, 92, 189, 239, 253, 402, 406, 424, 488, 493, 500

Comment 12

Line 33: comma after problematic unnecessary

Comment 13

Following sentences must be restructured as they are very difficult to follow: lines 38-40, 69-71, 177-179, 409-410

Comment 14

Line 422: Typo – “CHBISs”

Reviewer #2: This is clearly an important paper that outlines how a health activist group used health information to improve healthcare in a low-income area of Cape Town. I commend the authors for capturing the key strategies that were employed by the activist group to enact change. I have some comments that I hope would make the paper clearer and make the writing tighter. As a qualitative methodologist, I have focused on clarifying the methodology, ensuring the findings use quotes to demonstrate key themes that were derived from the interviews and following the reporting standards in checklists such as COREQ.

Background

Well written and covers key issues, but it would benefit from some restructuring to help the reader.

• For instance, it would be helpful, especially for those not familiar with the topic area, to explain what the authors mean by health information early in the background section. Health information is a broad term and it would be helpful to contextualise this to South Africa and provide an explanation of what it means in this article in relation to the WHO health system building blocks.

• Line 59 – ‘This paper investigates…’ – The authors could build up to the need for this rather than have this upfront – more so because the background section ends with this important bit in lines 123 -131 anyway. Instead, authors could introduce a brief explanation of what they mean by health information at the end of the first para, so that it leads nicely into the second para that provides more detailed info on health information.

• Line 82 – ‘some initiatives’ – where and by whom please? are these government initiatives?

• Line 86 – are CBHISs the norm in all of Africa except in S.Africa as indicated from line 96 onwards? If there are variations across the rest of Africa, useful to avoid ‘In Africa’ and clarify this in line 86 (even saying ‘In some countries in Africa’ or similar would help).

• Line 96 – are HFCs and clinic committees the same/similar? Apologies if I’ve got this wrong, but I’m assuming the key point made in this para is that while in some parts of Africa, there are CBHISs, in S.Africa this is not the case; so HFCs and clinic committees use other means to collect and use health information, although there isn’t much literature on this. That doesn’t come through clearly in this para, took a few readings - would really help to clarify.

• There are many abbreviations throughout the manuscript, which is fine, but may be useful to include a table with the abbreviations, their expansion and what they actually mean or do, as supplementary material, to provide context for anyone interested in looking this up further.

Methods

Kindly include a COREQ or similar checklist as an appendix (https://www.equator-network.org/reporting-guidelines-study-design/qualitative-research/). Including details in the manuscript to satisfy these criteria will also help address some of the key methodological queries raised. Please also strengthen the methods sections with appropriate references – there are none in submitted version.

• Lines 137 and 152 – repeats info but slightly different on short-term and long-term focus of MCSJ.

• Line 184 – instead of extended period of time, specify for how long please.

• Methods sections needs specific info as outlined in COREQ – for instance, were topic guides used for interviews, were interviews audio-recorded and transcribed, were interviews coded by more than one person, how many interviews/observations, etc.

• Please consider rephrasing 'themes emerged' ‘emerging themes’ in manuscript in line with current thinking in qualitative research that these phrases does not take into account researchers' roles/is passive, as opposed to themes being 'identified', which is considered a more active process (please see Braun and Clarke's work, which elaborates on this).

Findings and discussion

• Use of the term ‘Dentistry’ – sorry to be pedantic and this might be just down to differences in how we use these terms in different countries, so apologies if this is how it is used in S.Africa. As someone who used to be a dentist, to me, dentistry is usually used to mean dental medicine and not used to signify a place (where dental clinic is better) – like in line 233 and other places throughout manuscript.

• I understand methods, such as observations were used, but given this is a qualitative research paper that used semi-structured interviews, one would expect to see more quotations – there is just one used. Please ensure findings that draw from interview data are adequately demonstrated with quotes (see COREQ). Similarly, it would be helpful to know which findings were drawn from which data sources so that it is clear to the reader that the findings are grounded in the data collected.

• I found it somewhat difficult to capture the key messages within each of the strategies – they could do with a really tight edit overall so that the most important findings within each are clear to the readers. It was also difficult to make out which aspects were from the data and which were the authors views based on their experiences or existing literature (note that the latter is important too, but perhaps findings is not the section for it) – e.g. lines 266-275 and similarly throughout the findings section. This paper would really benefit by keeping the findings tight and directly linked to the data from this study, with other aspects moved to discussion.

• While it is important to reinforce some key messages, it would be important to not be repetitive – for instance, ‘social process rather than a technical exercise’ or similar is repeated three times in the discussion.

• Lines 460-461 – can’t see the ‘hunger for information’ mentioned in the findings. Ideal to not bring in new findings in the discussion, without mentioning them in the results first.

Ethical issues

• Line 198 – in what instances was verbal consent sought instead of written consent and why?

• Also, could you please indicate whether the people identified by name throughout the manuscript (activists, healthcare professionals, etc) have provided consent for the same.

• Readers are likely to be curious about the informed consent aspects of the data collection activities of MCSJ, such as the quantitative data from patients in the queue for the dental clinic and the qualitative testimonials from ‘patients living with HIV’ (also, request to use people-first language instead of ‘HIV+ patients’ where possible). How was anonymity/confidentiality operationalised, given this information was then shared widely in local groups? Were any ethical issues in relation to this identified in the interviews, etc.?

Minor typos, etc.

• Line 31: Could you please amend to ‘However, many health system stakeholders, including community groups, are perpetually…’

• Line 123 – expand HIS please (it’s within other abbreviations before, but worth expanding separately).

• Line 170 – ‘who were recruited’

• Line 172 – ‘when participating in observations of meetings’

• Line 208 – ‘various creative’

• Lines 443-444 – CBO or CTO?

Spotted a few more, but if paper is accepted these will be picked up in the proofreading process. For now, my view is that it is best to focus on tightening the paper so the messages are clear and ensure the methodological aspects are clear, with the inclusion of a COREQ checklist to help with that. Hope that helps – wishing you good luck with the paper, I enjoyed reading it.

6. PLOS authors have the option to publish the peer review history of their article (what does this mean?). If published, this will include your full peer review and any attached files.

**Do you want your identity to be public for this peer review?** For information about this choice, including consent withdrawal, please see our Privacy Policy.

Reviewer #1: No

Reviewer #2: No

---

## [Decision Letter · Decision Letter 1]

23 May 2022

PGPH-D-21-00746R1

Using health information for community activism: A case study of the Movement for Change and Social Justice in South Africa.

Dear Dr. van Pinxteren,

Thank you for submitting your manuscript to PLOS Global Public Health. After careful consideration, we feel that it has merit but does not fully meet PLOS Global Public Health’s publication criteria as it currently stands. Therefore, we invite you to submit a revised version of the manuscript that addresses the points raised during the review process.

Dear authors,

Your resubmission has now been reviewed by 2 reviewers, and Reviewer 1 is making the below comments recommending minor revision. If you can address , we can then move to accepting the paper.

We look forward to receiving your revised manuscript.

Kind regards,

Isabelle Uny

Academic Editor

Journal Requirements:

Additional Editor Comments (if provided):

Reviewers' comments:

Reviewer's Responses to Questions

**Comments to the Author**

1. If the authors have adequately addressed your comments raised in a previous round of review and you feel that this manuscript is now acceptable for publication, you may indicate that here to bypass the “Comments to the Author” section, enter your conflict of interest statement in the “Confidential to Editor” section, and submit your "Accept" recommendation.

Reviewer #1: (No Response)

Reviewer #2: (No Response)

2. Does this manuscript meet PLOS Global Public Health’s publication criteria? Is the manuscript technically sound, and do the data support the conclusions? The manuscript must describe methodologically and ethically rigorous research with conclusions that are appropriately drawn based on the data presented.

Reviewer #1: Yes

Reviewer #2: Partly

3. Has the statistical analysis been performed appropriately and rigorously?

Reviewer #1: Yes

Reviewer #2: N/A

4. Have the authors made all data underlying the findings in their manuscript fully available (please refer to the Data Availability Statement at the start of the manuscript PDF file)?

Reviewer #1: Yes

Reviewer #2: No

5. Is the manuscript presented in an intelligible fashion and written in standard English?

Reviewer #1: Yes

Reviewer #2: Yes

6. Review Comments to the Author

Reviewer #1: You have addressed all of my initial recommendations and I am happy to support publication. It is a very interesting paper!

Reviewer #2: Comments attached separately

7. PLOS authors have the option to publish the peer review history of their article (what does this mean?). If published, this will include your full peer review and any attached files.

**Do you want your identity to be public for this peer review?** For information about this choice, including consent withdrawal, please see our Privacy Policy.

Reviewer #1: No

Reviewer #2: No

---

## [Editor Report · Decision Letter 2]

18 Aug 2022

Using health information for community activism: A case study of the Movement for Change and Social Justice in South Africa.

PGPH-D-21-00746R2

Dear Dr van Pinxteren,

We are pleased to inform you that your manuscript 'Using health information for community activism: A case study of the Movement for Change and Social Justice in South Africa.' has been provisionally accepted for publication in PLOS Global Public Health.

Best regards,

Isabelle Uny

Academic Editor